# Suicidality among Chinese college students: A cross-sectional study across seven provinces

Bob Lew[1]⊕, Kairi Kõlves[2,3,4]⊕, Augustine Osman[5]⊕, Mansor Abu Talib[6]⊕, Norhayati Ibrahim[7]⊕, Ching Sin Siau [8]⊕*, Caryn Mei Hsien Chan[7]⊕

1 Department of Social Psychology, Faculty of Human Ecology, Putra University of Malaysia, Serdang, Selangor, Malaysia, 2 Australian Institute for Suicide Research and Prevention, Griffith University, Brisbane, Queensland, Australia, 3 WHO Collaborating Centre for Research and Training in Suicide Prevention, Griffith University, Brisbane, Queensland, Australia, 4 School of Applied Psychology, Griffith University, Brisbane, Queensland, Australia, 5 Department of Psychology, University of Texas at San Antonio, San Antonio, Texas United States of America, 6 Department of Human Development and Family Studies, Putra University of Malaysia, Serdang, Selangor, Malaysia, 7 Faculty of Health Sciences, Universiti Kebangsaan Malaysia, Kuala Lumpur, Malaysia, 8 Faculty of Social Sciences and Liberal Arts, UCSI University, Kuala Lumpur, Malaysia

⊕ These authors contributed equally to this work.
* chingsin.siau@gmail.com

**Data Availability Statement:** All relevant Data is available in the Manuscript and its Supporting Information files.

## Abstract

### Background

Although the suicide rate in China has decreased over the past 20 years, there have been reports that the younger age group has been experiencing an increased incidence of completed suicide. Given that undergraduate groups are at higher risks of suicidality, it is important to monitor and screen for risk factors for suicidal ideation and behaviors to ensure their well-being.

### Objective

To examine the risk and protective factors contributing to suicidality among undergraduate college students in seven provinces in China.

### Methods

We conducted a cross-sectional study involving 13,387 college students from seven universities in Ningxia, Shandong, Shanghai, Jilin, Qinghai, Shaanxi, and Xinjiang. Data were collected using self-report questionnaires.

### Results

Higher scores in the psychological strain, depression, anxiety, stress, and psychache (psychological risk factors for suicidality) and lower scores in self-esteem and purpose in life (psychological protective factors against suicidality) were associated with increased suicidality among undergraduate students in China. Demographic factors which were associated with higher risks of suicidality were female gender, younger age, bad academic results,

**Funding:** CCMH acknowledges support from the Universiti Kebangsaan Malaysia grant DIP-2018-035. The funders had no role in study design, data collection and analysis, decision to publish, or preparation of the manuscript.

**Competing interests:** The authors have declared that no competing interests exist.

were an only child, non-participation in school associations, and had an urban household registration. Perceived good health was protective against suicidality.

## Conclusions

Knowing the common risk and protective factors for suicidality among Chinese undergraduate students is useful in developing interventions targeted at this population and to guide public health policies on suicide in China.

## Introduction

The World Health Organization estimated about 793,000 suicide deaths worldwide in 2016, and a global age-standardized suicide rate of 10.5 per 100,000 population [1]. There are approximately 25 suicide attempts for every suicide death, and one suicide death is estimated to affect 135 people [2], resulting in 108 million people world-wide being negatively impacted by suicide.

Although China is the second largest economy in the world (with a Gross Domestic Product (GDP) of $13.6 trillion in 2018, compared to the United States at $20.5 trillion) [3], its suicide rate has decreased in the past 20 years from above 20/100,000 population to the current rate of 8/100,000 population [4]. The current rate translates to an estimated 112,000 deaths, 2.8 million attempted suicides, and 15.1 million people impacted by suicide in China [1]. This rate is lower than in the US which is estimated to be 13.7/100,000, and ranks between the suicide rates of the world's ten largest economies which range from 5.5/100,000 (Italy) to 16.5/100,000 (India) [1]. Zhang has outlined that this drop may possibly be due to the following six factors: (1) fast economic development; (2) migration to urbanised areas; (3) modernised social values; (4) one-child per family policy; (5) college surveillance-based counselling; and (6) governmental media control [4].

Suicide is widely recognised as the second leading cause of death in young people in the 15- to 29-years age bracket worldwide [1]. There have been reports that this younger age group has been experiencing an increased incidence of completed suicide in China [5,6]. Thus, it is important to focus on individuals within this age bracket as they still have a long life-cycle. The emotional as well as the economic costs incurred by the suicide deaths of this age group are high [7]. Most undergraduate students fall within the 18 to 29 age range. Given that undergraduate groups are the emerging generation which determine China's future, it is important to monitor and screen for risk factors for suicidal ideation and behaviors to ensure their well-being.

There is a collective body of evidence on the identification of risk and protective factors for suicide, which includes factors such as life stress and coping style [8], personality factors of impulsivity and aggression [9], and depression [9–12]. In addition, adverse developmental influences or events such as childhood adversity, divorce of parents, loss of a parent, and sexual abuse are also risk factors for increased suicidality [9–12]. Among Chinese college students, factors that have been found to influence suicidality are academic performance, academic stress, occupational future, recent conflicts with classmates, satisfaction with major, and the rupture of romantic relationships [12,13]. Finally, family influences such as parents' educational level, family income, a history of suicide in the family, and originating from a rural background are also factors that are associated with suicidality among college students [14–19].

Nevertheless, there still remains a lack of comprehensive studies on various common suicidality risk and protective factors on this population segment in China using sufficiently large samples. Although it may not be possible to study all colleges and provinces in a large country like China simultaneously, this research attempts to address this gap, using established and relevant psychological measurements. To the authors' best knowledge, this is the largest study undertaken on risk and protective factors for suicide among undergraduate students in China, spanning seven provinces with more than 13,000 samples. A further understanding of risk and protective factors for suicide is needed to guide public health policies on suicide in China.

## Materials and methods

### Study design

We conducted a large cross-sectional cluster sample study of undergraduate college students in seven provinces in China comprising Ningxia, Shandong, Shanghai, Jilin, Qinghai, Shaanxi, and Xinjiang.

### Data collection

Data was collected in survey format from students enrolled in various undergraduate degree programs from universities in seven provinces. One university was selected in each province. Students from each department were clustered according to the year of study. An equal number of classes from each year of study were then selected to obtain a reasonable representation of each grade. All students in the selected classes were briefed about the purpose of the research. Participants completed the questionnaires anonymously in approximately a half an hour, and no identifiers were collected. This study received ethics approval from the institutional review board of the Ethics Committee at the School of Public Health, Shandong University (No. 20161103). The participant signed an informed consent form before answering the questionnaire. Hotlines on counseling services were provided in the information sheet tailored to each province. Exclusion criteria were determined a priori as follows: 1) if gender and age were not provided; and 2) if all items on the Suicidal Behaviors Questionnaire-Revised were not completed.

### Measurements

**Demographics.** The following information was collected: age (using year of birth), gender (male = 1; female = 2), physical health (poor = 1 to good = 3), economic status (poor = 1 to good = 3), academic results (poor = 1 to good = 3), only child status (yes = 1; no = 2), participation in school associations (yes = 1; no = 2), and household registration (urban = 1; rural = 2).

**Suicidal Behaviors Questionnaire-Revised (SBQ-R).** The SBQ-R was developed as a brief measure of a range of suicide-related behaviors for use in both clinical and nonclinical settings. It is a 4-item self-report questionnaire [20]. The total score ranges from 3 to 18, with higher scores indicating greater risk of suicidal behaviors. A cut-off score of $\geq 7$ is used to indicate suicide risk for undergraduate students [20]. Lifetime suicidal ideation and lifetime suicide attempt were determined by the question, "Have you ever seriously thought about suicide?". Participants who responded "It was just a brief passing thought" indicated lifetime suicidal ideation. Participants who responded "I have attempted to kill myself, but did not want to die" or "I have attempted to kill myself, and really hoped to die" indicated lifetime suicide attempt. The Chinese translated version yielded an internal consistency estimate of Cronbach's α = 0.67 [21]. In this study, the Cronbach's α of the scale score is 0.75.

### Risk factors

**DASS-21.** DASS-21 is a well-established instrument comprising three dimensions of psychological distress, including depression, anxiety and stress [22]. Each dimension is measured by seven items. The score on each item ranges from 0 = "did not apply to me at all" to 3 = "applied to me very much", or "most of the time"). The total score ranges from 0 to 63. DASS-21 has been widely used in China for various psychosocial studies, such as Cheng et al. which reported an internal consistency estimate of Cronbach's α = 0.77 [23]. In this study, the Cronbach's α of the scale score is 0.95.

**Psychache scale.** The Psychache Scale [24] consists of 13 items reflecting psychache (i.e., mental pain) scored from 1 = "never or strongly disagree" to 5 = "always or strongly agree". The Psychache Scale scores have adequate psychometric properties, with alpha reliability coefficients over 0.90 when completed by university students. In a sample of Chinese students, the Cronbach's α was 0.94 [21]. In this study, the Cronbach's α of the scale score is 0.96.

**Psychological strain.** The Psychological Strain Scales (PSS), which showed good validity and reliability estimates both in Chinese and American college samples, were first developed in a Chinese sample to measure the level of strain [25]. The PSS consists of the four dimensions of psychological strains, namely, value strain, aspiration strain, deprivation strain, and coping strain; each of these dimensions contains 10 items. For example, "I am often confused about what life means to me" corresponds to value strain; "I wish I had a better job now, but I cannot realize it according to some reasons" corresponds to aspiration strain; "Compared to others in my neighborhood (village), I am a poor person" corresponds to deprivation strain; and "Face is so important to me that I will do everything to protect my public image, even suicide" corresponds to coping strain. Response options for each item were as follows: 1 = never/not me at all; 2 = rarely/not me; 3 = maybe/not sure; 4 = often/like me; 5 = yes, strongly agree/exactly like me. The total score for each of the four strains was obtained by summing the total score of each dimension (10 items). The higher the total score of the PSS was, the greater the level of psychological strains. In this study, the Cronbach's α of the scale score is 0.96.

### Protective factors

**Self-esteem scale.** The self-esteem scale (SES), developed by Rosenberg [26], was originally used to assess adolescents' overall feelings of self-worth and self-acceptance, is currently the most widely used self-esteem measure of this construct, and has solid reliability estimate (Cronbach's α = 0.84). Although commonly used among adolescents, the scale has also been used in a Chinese college student sample.[27] The SES consists of 10 items, with every item ranging from 1 = "strongly disagree" to 4 = "strongly agree". The total score ranges from 10 to 40. Higher scores indicate greater self-esteem. The Chinese SES was tested in China and had acceptable reliability estimate (Cronbach's α = 0.71) [27]. In this study, the Cronbach's α of the scale score is 0.66.

**Purpose in life.** The four-item purpose in life test–short form (PLT-SF) was used to measure the extent to which participants felt their lives had meaning and purpose [28]. The PLT-SF includes a 7-point Likert-type scale response format. Responses to the items are summed to obtain a total score ranging from 4 to 28. Higher scores indicate greater perceived meaning/purpose in life. A Chinese version of the PLT-SF has been shown to have acceptable reliability estimate (Cronbach's α = 0.89) [29]. In this study, the Cronbach's α of the scale score is 0.92.

### Statistical analysis

Univariate and multivariate statistical analyses were conducted using the IBM SPSS v.21 (SPSS Inc.; Armonk, NY). The participant demographics were computed using descriptive statistics.

Independent samples $t$-tests and one-way analyses of variance were conducted to test for significant mean differences between age groups, gender, economic status, academic results, only child status, participation in school associations, household registration, year of study, race, other province, and political party membership. A multiple regression analysis was used to determine the significant predictors of suicide risk based on SBQ-R total scores which were log natural transformed. A binary logistic regression was conducted to test the model for the odd ratios for being at risk of suicide (SBQ-R total score $\geq$ 7). For all comparisons, differences were determined using two-tailed tests while $p$-values less than 0.05 were considered statistically significant. Missing data were deleted list-wise during the statistical analysis.

## Results

### Participants

The total number of participants from seven provinces was 13,387. After deleting missing and out of range data for demographics and the SBQ-R, 11,473 participants were included in the final analysis (mean age = 20.69±1.35).

### Demographic factors

**Age.** The highest suicidality score was reported by the 19-year-old (y.o.) age group at 1.44 ±0.42 and the 24 y.o. age group reported the lowest suicidality at 1.33±0.37 which is significantly lower than all the other age groups, $F_{(6, 11,472)}$ = 4.53, $p<0.001$.

**Gender.** Female students reported higher suicidality with a mean SBQ-R total score of 1.44 ±0.43 vs. male students with 1.40±0.41, $t_{(9957.61)}$ = -4.61, $p<0.001$.

**Physical health.** Students who self-reported "poor" physical health scored the highest in suicidality at 1.55±0.49 compared to those who reported "normal" and "good" physical health, $F_{(2, 11,468)}$ = 135.19, p<0.001.

**Economic status.** Students who self-reported that they have "poor" economic status scored the highest at 1.46±0.45 followed by "good" and "normal", $F_{(2, 11,461)}$ = 14.09, $p<0.001$.

**Academic results.** Those who reported "poor" academic results had the highest level of suicidality mean score at 1.57±0.48, followed by "normal" and "good", $F_{(2, 11,454)}$ = 62.11, $p<0.001$. The "good" category scored the lowest among the three categories ($p<0.001$). There appears to be a trend effect of the three categories of academic results (i.e. "poor" >"normal" > "good") on suicidal behaviors.

**Only child.** Students who reported that they come from an only child family reported a higher suicidality mean score at 1.44±0.43 than students who reported that they have siblings at 1.41±0.42, $t_{(7561.70)}$ = 3.79, $p<0.001$.

**School associations.** Students who reported that they participated in school associations reported a lower suicidality mean score at 1.42±0.42 than those who do not participate in school activities 1.44±0.43, $t_{(5371.37)}$ = -2.06, $p = 0.039$.

**Household registration.** Students with an urban household registration reported a higher suicidality mean score at 1.46±0.44 compared to students who reported a rural household registration of 1.39±0.40, $t_{(11,048.17)}$ = 7.95, $p<0.001$ (Table 1).

### Risk and protective factors for suicidality

The results of the multiple linear regression indicated that psychache ($\beta = 0.162$, $p < 0.001$) and DASS-21 ($\beta = 0.155$, $p < 0.001$) demonstrated the strongest associations with increased suicidality. Together, the independent variables accounted for 22.4% of the variance, $R^2 = 0.221$, adjusted $R^2 = 0.219$, $F_{(22, 10,455)}$ = 134.32, $p<0.001$ (Table 2).

**Table 1. Mean comparison of demographic variables with suicidality (SBQ-R total score) (N = 11,473).**

| Demographic Variables | N | % | Mean±SD | *t / F* statistics | Post-Hoc and *p*-value |
|---|---|---|---|---|---|
| Age | | | | 4.53*** | |
| 18 (1) | 225 | 2.0 | 1.432±.406 | | 1>7 *p* = .004 |
| 19 (2) | 2058 | 17.9 | 1.438±.423 | | 2>5 *p* = .026 |
| 20 (3) | 3313 | 28.9 | 1.435±.426 | | 2>7 *p* < .001 |
| 21 (4) | 2928 | 25.5 | 1.420±.424 | | 3>5 *p* = .025 |
| 22 (5) | 1707 | 14.9 | 1.407±.411 | | 3>7 *p* < .001 |
| 23 (6) | 895 | 7.8 | 1.405±.426 | | 4>7 *p* < .001 |
| 24 (7) | 347 | 3.0 | 1.329±.370 | | 5>7 *p* = .001 |
| | | | | | 6>7 *p* = .004 |
| Gender | | | | -4.61*** | 2>1 *p* < .001 |
| Male (1) | 4496 | 39.2 | 1.400±.407 | | |
| Female (2) | 6977 | 59.8 | 1.436±.430 | | |
| Physical Health | | | | 135.19*** | 1>2 *p* = 0.002 |
| Poor (1) | 740 | 6.4 | 1.554± 0.487 | | 1>3 *p*<0.001 |
| Normal (2) | 3749 | 32.7 | 1.489± 0.451 | | |
| Good (3) | 6980 | 60.8 | 1.372± 0.388 | | |
| Missing | 4 | | | | |
| Economic Status | | | | 14.09*** | 1>2 *p*<0.001 |
| Poor (1) | 2398 | 20.9 | 1.458± | | 3>2 *p* = 0.038 |
| Normal (2) | 7599 | 66.2 | 1.408± | | |
| Good (3) | 1465 | 12.8 | 1.437± | | |
| Missing | 11 | | | | |
| Academic Results | | | | 62.11*** | 1>2,3 *p* < .001 |
| Poor (1) | 882 | 7.7 | 1.571±480 | | |
| Normal (2) | 8318 | 72.6 | 1.413±413 | | |
| Good (3) | 2255 | 19.7 | 1.397±413 | | |
| Missing | 18 | | | | |
| Only Child | | | | 3.79*** | 1>2 *p* < .001 |
| Yes (1) | 3868 | 33.9 | 1.443±.431 | | |
| No (2) | 7537 | 66.1 | 1.411±.416 | | |
| Missing | 68 | | | | |
| School Associations | | | | 2.06* | 2>1 *p* = .039 |
| Yes (1) | 8358 | 73.0 | 1.417±.418 | | |
| No (2) | 3094 | 27.0 | 1.436±.431 | | |
| Missing | 21 | | | | |
| Household Registration | | | | 7.95*** | 1>2 *p* < .001 |
| Yes (1) | 5419 | 47.3 | 1.455±.438 | | |
| No (2) | 6026 | 52.7 | 1.392±.403 | | |
| Missing | 28 | | | | |

*p<0.05;

**p<0.01;

***p<0.001. Games-Howell post-hoc analysis was employed.

**Table 2. Multiple linear regression of factors associated with suicidality (SBQ-R total score).**

| Variables | B | B SE | 95% CI | | β | t | p-value |
|---|---|---|---|---|---|---|---|
| | | | **Upper Bound** | **Lower Bound** | | | |
| Psychological Strain | 0.001 | 0.000 | 0.001 | 0.002 | 0.077 | 6.542 | <0.001 |
| DASS-21 | 0.005 | 0.000 | 0.004 | 0.006 | 0.155 | 11.226 | <0.001 |
| Psychache | 0.007 | 0.001 | 0.006 | 0.008 | 0.162 | 12.045 | <0.001 |
| Self-Esteem | -0.005 | 0.001 | -0.008 | -0.003 | -0.045 | -4.201 | <0.001 |
| Purpose-in-Life | -0.009 | 0.001 | -0.011 | -0.007 | -0.097 | -9.389 | <0.001 |
| Province | | | | | | | |
| Jilin* | | | | | | | |
| Ningxia | -0.015 | 0.013 | -0.041 | 0.011 | -0.013 | -1.121 | 0.262 |
| Shandong | -0.081 | 0.014 | -0.108 | -0.054 | -0.062 | -5.879 | <0.001 |
| Shanghai | 0.032 | 0.013 | 0.006 | 0.058 | 0.028 | 2.408 | 0.016 |
| Qinghai | -0.094 | 0.015 | -0.123 | -0.065 | -0.068 | -6.330 | <0.001 |
| Shaanxi | -0.083 | 0.013 | -0.109 | -0.057 | -0.074 | -6.283 | <0.001 |
| Xinjiang | -0.096 | 0.014 | -0.124 | -0.068 | -0.073 | -6.685 | <0.001 |
| Age | -0.012 | 0.003 | -0.018 | -0.007 | -0.039 | -4.320 | <0.001 |
| Gender | | | | | | | |
| Male* | | | | | | | |
| Female | 0.077 | 0.008 | 0.062 | 0.092 | 0.089 | 9.871 | <0.001 |
| Physical Health | | | | | | | |
| Normal* | | | | | | | |
| Good | -0.032 | 0.008 | -0.048 | -0.016 | -0.037 | -3.879 | <0.001 |
| Poor | -0.027 | 0.016 | -0.059 | 0.005 | -0.016 | -1.668 | 0.095 |
| Economic Status | | | | | | | |
| Normal* | | | | | | | |
| Good | 0.029 | 0.012 | 0.006 | 0.053 | 0.023 | 2.452 | 0.014 |
| Poor | 0.001 | 0.010 | -0.018 | 0.020 | 0.001 | 0.058 | 0.954 |
| Academic Results | | | | | | | |
| Normal* | | | | | | | |
| Good | 0.011 | 0.010 | -0.008 | 0.030 | 0.011 | 1.156 | 0.248 |
| Poor | 0.080 | 0.014 | 0.051 | 0.108 | 0.050 | 5.503 | <0.001 |
| Only Child | | | | | | | |
| No* | | | | | | | |
| Yes | 0.021 | 0.008 | 0.004 | 0.037 | 0.023 | 2.479 | 0.013 |
| School Associations | | | | | | | |
| No* | | | | | | | |
| Yes | 0.026 | 0.008 | -0.043 | 0.010 | 0.043 | 3.094 | 0.002 |
| Household Registration | | | | | | | |
| Rural* | | | | | | | |
| Urban | 0.046 | 0.008 | 0.030 | 0.062 | 0.054 | 5.577 | <0.001 |

*Reference group.

The results of the binary logistic regression indicated that participants who scored higher in psychological strain, DASS-21 and psychache were at an increased risk of suicidality ($p<0.001$). Meanwhile, those with higher scores in self-esteem ($p = 0.001$) and purpose in life ($p<0.001$) were at a decreased risk of suicidality. In terms of demographic factors, females,

those with perceived good economic status, poor academic results, were an only child, who participated in school associations, and had an urban household registration were at an increased risk of suicidality ($p<0.001$ to $p<0.05$). Meanwhile, perceived good health was protective against suicidality (Table 3).

**Table 3. Binary logistic regression of factors associated with suicidality (SBQ-R total score).**

| Variables | Odds Ratio | 95% CI | | p-value |
|---|---|---|---|---|
| | | Lower | Upper | |
| Psychological Strain | 1.009 | 1.006 | 1.012 | <0.001 |
| DASS-21 | 1.033 | 1.026 | 1.040 | <0.001 |
| Psychache | 1.036 | 1.028 | 1.044 | <0.001 |
| Self-Esteem | 0.968 | 0.949 | 0.987 | 0.001 |
| Purpose-in-Life | 0.947 | 0.935 | 0.960 | <0.001 |
| Province | | | | |
| Jilin* | | | | |
| Ningxia | 1.313 | 1.066 | 1.618 | 0.010 |
| Shandong | 0.931 | 0.747 | 1.160 | 0.524 |
| Shanghai | 1.418 | 1.175 | 1.711 | <0.001 |
| Qinghai | 0.898 | 0.719 | 1.122 | 0.344 |
| Shaanxi | 0.798 | 0.653 | 0.976 | 0.028 |
| Xinjiang | 0.686 | 0.527 | 0.891 | 0.005 |
| Age | 0.930 | 0.890 | 0.973 | 0.001 |
| Gender | | | | |
| Male* | | | | |
| Female | 1.679 | 1.484 | 1.898 | <0.001 |
| Physical Health | | | | |
| Normal* | | | | |
| Good | 0.773 | 0.683 | 0.874 | <0.001 |
| Bad | 0.867 | 0.700 | 1.074 | 0.191 |
| Economic Status | | | | |
| Normal* | | | | |
| Good | 1.240 | 1.036 | 1.486 | 0.019 |
| Bad | 1.003 | 0.868 | 1.158 | 0.968 |
| Academic Results | | | | |
| Normal* | | | | |
| Good | 1.037 | 0.888 | 1.211 | 0.648 |
| Bad | 1.306 | 1.074 | 1.589 | 0.008 |
| Only Child | | | | |
| No* | | | | |
| Yes | 1.192 | 1.052 | 1.351 | 0.006 |
| School Associations | | | | |
| No* | | | | |
| Yes | 1.231 | 1.081 | 1.401 | 0.002 |
| Household Registration | | | | |
| Rural* | | | | |
| Urban | 1.265 | 1.117 | 1.432 | <0.001 |

*Reference group.

## Discussion

Our findings indicated that higher scores in the psychological risk factors (psychological strain, depression, anxiety, stress, and psychache) and lower scores in the protective factors for suicidality (self-esteem and purpose in life) were associated with increased suicidality among undergraduate students in China. This is similar to the results of a number of studies carried out in the West [30–33]. However, the "negative life events" experienced by Chinese undergraduate students may differ, given that China has a different socio-political environment, educational approach, and even campus administration style [4].

In terms of demographic factors, females were at an increased suicide risk, a finding which is replicated in a number of Western and Chinese studies on college students [15 34–36]. This may be due a number of factors, including a brooding ruminative style among females [35] or wider sociocultural issues affecting females such as gender inequality, especially among female individuals who adhere to Confucian ethics [37–39]. Younger college students may be more susceptible to developing suicidality as they may still be transitioning to college life [40]. Perceived good physical health was protective of suicide. The late adolescence and young adulthood are considered the healthiest periods of an individual's life [41]. Therefore, perceived poor health could be very distressing for this age group.

Undergraduates with an urban household registration reported higher suicidality compared to those with a rural household registration. This is in contrast with an earlier study which indicated no difference in the suicidality levels between rural and urban Chinese college students [42], or higher levels of suicidality among rural Chinese college students [10]. The findings may reflect the narrowing of rural:urban ratios in suicide rate, which has traditionally been higher among the rural Chinese [43].

Another interesting finding is that participants who reported a good economic status were at an increased risk of suicidality, as were those who participated in school activities. These results are inconsistent with past findings, as past studies have indicated that students from a lower socioeconomic background reported higher suicidality levels [44]. Participation in school activities was seen as being protective of suicidality because exposure to a wider social circle may increase social support [45]. Further investigations need to be conducted to test the possible mediating variables in the relationship between socioeconomic background and school activities with suicidality.

In the past 20 years, a decrease of suicide rates by 43% in China from 14.1 in 2000 to 8.1 per 100,000 population in 2016 has been recorded [46]. Compared to the US, college students in China have reported lower suicide-related behaviors risk [47]. The public health implementation of targeted suicide prevention activities may have worked synergistically to lower the suicide rates, apart from the possible effects of socioeconomic development. Examples include regulatory changes in the use of pesticides, or the implementation of the lock-box method for safe pesticide storage [48], the success of which was facilitated by urbanisation and migration from rural areas to the cities, where there is less access to pesticides. The success of such means restriction policies has also been successfully implemented in other Lower- and Middle-Income Countries such as India [49], and may inform firearm regulation policies in the US [50].

## Conclusion

Understanding the common risk and protective factors for suicide among Chinese undergraduate college students is crucial in building a resilient and appropriate suicide prevention programme for this segment of society. The identified protective factors such as self-esteem and

purpose in life could be reinforced through various suicide prevention and character-building curricula. Early intervention on this population could have long-term positive consequences.

## Limitations

This is a cross-sectional study across seven provinces, thus causality could not be inferred. Provincial comparisons were not conducted as only one urban university was selected for each province. The results are therefore not representative of Chinese college students. There may be other important risk and protective factors for suicide which are not included in this study due to questionnaire length constraints. For example, what students do during their free time was not taken into consideration. The additional stress from working or caring for another individual may affect their emotional health compared to those who do not have such activities. In addition, the time of administering survey should be considered as specific times during the semester (e.g. before exams) may be associated with more stress and negative emotions. Therefore, future studies should include other risk factors facing college students such as parental and other relationships and personal time activities. The identification of stressful timepoints would enable school authorities to assist with timing for implementing more suicide intervention activities. A representative sampling of Chinese universities should be undertaken in the future to ensure the wider generalizability of the results.

## Supporting information

**S1 Dataset. PLoSOne student suicidality dataset.**
(SAV)

## Acknowledgments

The authors would like to thank the seven collaborators and associates in China for their coordination in data collection for each province: Prof. Cai Honglan (The People Hospital of Qinghai Province, Qinghai Minzu College, Qinghai Province), Prof. Jia Cunxian (Department of Epidemiology, School of Public Health, Shandong University; Shandong University Center for Suicide Prevention Research, Shandong Province), Prof. Kou Changgui (Department of Epidemiology & Biostatistics, School of Public Health, Jilin University, Jilin Province), Prof. Liu Changlin (Shanghai University), Prof. Liu Jiamin (Xinjiang Medical University, Xinjiang Autonomous Region), Prof. Liu Qiling (Department of Epidemiology and Health Statistics, School of Public Health, Shaanxi University of Chinese Medicine, Shaanxi Province), and Prof. Wang Zhizhong (Department of Epidemiology and Health Statistics, School of Public Health, Ningxia Medical University, Ningxia Autonomous Region). The authors also wish to thank the lecturers, post-graduate students, and the participants involved in this study. We wish to thank Prof. Wang Zhizhong who participated in the research design and the data coordination work in China.

## Author Contributions

**Conceptualization:** Bob Lew, Augustine Osman.

**Data curation:** Bob Lew.

**Formal analysis:** Bob Lew, Augustine Osman, Ching Sin Siau, Caryn Mei Hsien Chan.

**Investigation:** Bob Lew.

**Methodology:** Bob Lew, Augustine Osman, Ching Sin Siau, Caryn Mei Hsien Chan.

**Project administration:** Bob Lew.

**Resources:** Bob Lew.

**Supervision:** Mansor Abu Talib.

**Writing – original draft:** Bob Lew, Augustine Osman, Mansor Abu Talib, Ching Sin Siau, Caryn Mei Hsien Chan.

**Writing – review & editing:** Bob Lew, Kairi Kõlves, Augustine Osman, Mansor Abu Talib, Norhayati Ibrahim, Ching Sin Siau, Caryn Mei Hsien Chan.

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
