## [Decision Letter · Decision Letter 0]

11 May 2020

PONE-D-20-00103

Suicidality among Chinese college students: A cross-sectional study across seven provinces

PLOS ONE

Dear Dr. Siau,

Thank you for submitting your manuscript to PLOS ONE. After careful consideration, we feel that it has merit but does not fully meet PLOS ONE’s publication criteria as it currently stands. Therefore, we invite you to submit a revised version of the manuscript that addresses the points raised during the review process.

Please be advised that submitting a revised paper does not guarantee acceptance.

We would appreciate receiving your revised manuscript by Jun 25 2020 11:59PM. To enhance the reproducibility of your results, we recommend that if applicable you deposit your laboratory protocols in protocols.io, where a protocol can be assigned its own identifier (DOI) such that it can be cited independently in the future. For instructions see: http://journals.plos.org/plosone/s/submission-guidelines#loc-laboratory-protocols

We look forward to receiving your revised manuscript.

Kind regards,

Vincenzo De Luca

Academic Editor

PLOS ONE

Reviewers' comments:

Reviewer's Responses to Questions

**Comments to the Author**

1. Is the manuscript technically sound, and do the data support the conclusions?

Reviewer #1: Yes

Reviewer #2: Yes

2. Has the statistical analysis been performed appropriately and rigorously? 

Reviewer #1: I Don't Know

Reviewer #2: Yes

3. Have the authors made all data underlying the findings in their manuscript fully available?

Reviewer #1: Yes

Reviewer #2: No

4. Is the manuscript presented in an intelligible fashion and written in standard English?

Reviewer #1: Yes

Reviewer #2: Yes

5. Review Comments to the Author

Reviewer #1: Suicidality among Chinese college students: A cross-sectional study across seven provinces – REVIEW

Minor revisions required

The paper is well-written and contains a few minor grammar and spelling mistakes. The following are the line numbers and the mistake that should be fixed.

99 Minor error with the words ‘may be’.

As per review the methodology, results and discussion were well-written and explained clearly.

The methods should be revised slightly to create potentially a new category reviewing student personal time. This could be measured as no time = 1 and having time to rest/to themselves = 3. I believe this could increase the integrity of this study by emphasizing the importance of resting and having time for students to recover from various stressors.

The discussion honed into specific points and did not leave any gaps for other researchers to identify.

With regards to the discussion, I advise that additional information should be provided regarding China’s current public health guides on suicide. This is to provide researchers insight on what the current standards are in the country, and how results of this study compare to the processes being done. By providing this information, it could lead to potentially improving the current guides set in China, and a higher reduction in suicide rates in the coming years. It will also strengthen the study by exhibiting what country has done to successfully have a low rate of suicide in comparison to the United States. Through discussing this information researchers can therefore learn and set goals to improve not only China’s suicide rates, but also find solutions to use on the United States’ suicide health guides.

Some changes I recommend when revising this study, are to emphasize two other limitations that affected this study. One of them being the time this study took place, and the other with regards to the student’s personal life (i.e. do they work part-time, are they helping a family member etc.).

The time of the study was not emphasized and was not included in the paper. I recommend that a timeline on when students took the questionnaire be included, and then later mentioned as a potential factor. I believe time may be a variable and or limitation as specific times during semesters can lead to more stress, or negative emotions. It also should be mentioned as it gives other researchers reading this study the potential to identify if specific times in a year lead to higher suicide ideation, and suicide attempts.

What students do on their personal time should be taken in consideration or mentioned as a limitation that was not identified/studied. This is because additional stress from working, caring for another may affect their emotional health more than other students. Researchers may identify that this was missed in the study and could argue that female students may have a stronger association to suicidality due to cultural expectations (i.e. female Chinese students may have a stronger obligation to take care of their family than male Chinese students).

Reviewer #2: Thank you for asking me to review this article.

It clearly concerns an important and interesting topic, and is a useful piece of work. In that respect, I would support its publication. However, there are some areas that need resolving before publication.

1. The results as presented in the paper and the abstract do not agree; is participation in school associations a risk factor for suicidal ideation or not?

2. Statistically, I am not quite sure why the authours have chosen to impose a cut-off on the scoring level to determine suicidal ideation or not. I would have thought that it would be more sensible to assess the impact of potential factors on the outcome as a continuous variable. While I do not object to the more simple binary approach used here, it would be good to see a richer approach, treating the SBQ-R as a continuous variable.

3. Some of the risk factors could also be modelled better. For example, when considering Academic Results: "Academic Results. Those who reported “poor” academic results had the highest level of

231 suicidality mean score at 1.57±0.48, followed by “normal” and “good”, F (2, 11,454) = 62.11,

232 p<0.001. The “good” category scored the lowest among the three categories (p<0.001)."

This would seem to suggest that there is a trend effect across the three categories (i.e. Poor > Normal > Good). If that is the case, then please say so, and please consider modelling it.

4. The section on Data Sharing is unclear. THe submission states that it is available in the additional material, but that was not made available for review

5. The Ethics statement in the overview is insufficient; if there was appropriate IRB and individual level consent obtained (as appears to be the case) then this should be stated.

Based on this, my recommendation would be to revise and resubmit. I would be happy to review a revised version of the article.

Dr. M. Williams

Imperial College London

6. PLOS authors have the option to publish the peer review history of their article (what does this mean?). If published, this will include your full peer review and any attached files.

Reviewer #1: No

Reviewer #2: Yes: Dr. Matt Williams

---

## [Author Response · Author response to Decision Letter 0]

12 Jun 2020

Reviewer 1

Reviewer #1: Suicidality among Chinese college students: A cross-sectional study across seven provinces – REVIEW - Minor revisions required.

The paper is well-written and contains a few minor grammar and spelling mistakes. The following are the line numbers and the mistake that should be fixed.

1. 99 Minor error with the words ‘may be’.

Response: Thank you, the error has been corrected.

2. As per review the methodology, results and discussion were well-written and explained clearly.

The methods should be revised slightly to create potentially a new category reviewing student personal time. This could be measured as no time = 1 and having time to rest/to themselves = 3. I believe this could increase the integrity of this study by emphasizing the importance of resting and having time for students to recover from various stressors.

Response: Thank you for the suggestion. Unfortunately, we did not collect data on student personal/free time. However, we have included information on participation in school associations, which somewhat addresses what students do when not engaged in academic activities. We have addressed this in the limitations section, and will include the suggested variable in future studies that we carry out (refer response to Comment #4).

3. The discussion honed into specific points and did not leave any gaps for other researchers to identify.

With regards to the discussion, I advise that additional information should be provided regarding China’s current public health guides on suicide. This is to provide researchers insight on what the current standards are in the country, and how results of this study compare to the processes being done. By providing this information, it could lead to potentially improving the current guides set in China, and a higher reduction in suicide rates in the coming years. It will also strengthen the study by exhibiting what country has done to successfully have a low rate of suicide in comparison to the United States. Through discussing this information researchers can therefore learn and set goals to improve not only China’s suicide rates, but also find solutions to use on the United States’ suicide health guides.

Response: We have included newly added literature and references on the decline in suicide rates in China in the past 20 years, and compared the lower suicide behaviors risk of Chinese students with their counterparts in the US. China’s public health guides on suicide as a possible contributory factor has been added, as follows (ll. 303-313):

In the past 20 years, a decrease of suicide rates by 43% in China from 14.1 in 2000 to 8.1 per 100,000 population in 2016 has been recorded [4546]. Compared to the US, college students in China have reported lower suicide-related behaviors risk [47]. The public health implementation of targeted suicide prevention activities may have worked synergistically to lower the suicide rates, apart from the possible effects of socioeconomic development. Examples include regulatory changes in the use of pesticides, or the implementation of the lock-box method for safe pesticide storage [48], the success of which was facilitated by urbanisation and migration from rural areas to the cities, where there is less access to pesticides. The success of such means restriction policies has also been successfully implemented in other Lower- and Middle-Income Countries such as India [49], and may inform firearm regulation policies in the US [50].

4. Some changes I recommend when revising this study, are to emphasize two other limitations that affected this study. One of them being the time this study took place, and the other with regards to the student’s personal life (i.e. do they work part-time, are they helping a family member etc.).

The time of the study was not emphasized and was not included in the paper. I recommend that a timeline on when students took the questionnaire be included, and then later mentioned as a potential factor. I believe time may be a variable and or limitation as specific times during semesters can lead to more stress, or negative emotions. It also should be mentioned as it gives other researchers reading this study the potential to identify if specific times in a year lead to higher suicide ideation, and suicide attempts.

What students do on their personal time should be taken in consideration or mentioned as a limitation that was not identified/studied. This is because additional stress from working, caring for another may affect their emotional health more than other students. Researchers may identify that this was missed in the study and could argue that female students may have a stronger association to suicidality due to cultural expectations (i.e. female Chinese students may have a stronger obligation to take care of their family than male Chinese students).

Response: We have added the following limitations of the study in the revised manuscript (ll. 326-336):

There may be other important risk and protective factors for suicide which are not included in this study due to questionnaire length constraints. For example, what students do during their free time was not taken into consideration. The additional stress from working or caring for another individual may affect their emotional health compared to those who do not have such activities. In addition, the time of administering survey should be considered as specific times during the semester (e.g. before exams) may be associated with more stress, and negative emotions. Therefore, future studies should include other risk factors facing college students such as parental and other relationships and personal time activities. The identification of stressful timepoints would enable school authorities to assist with timing for implementing more suicide intervention activities.

Reviewer 2

Reviewer #2: Thank you for asking me to review this article.

It clearly concerns an important and interesting topic, and is a useful piece of work. In that respect, I would support its publication. However, there are some areas that need resolving before publication.

1. The results as presented in the paper and the abstract do not agree; is participation in school associations a risk factor for suicidal ideation or not?

Response: We apologise for the discrepancy, and thank you for pointing it out. Non-participation in school associations was a risk factor for suicidality, as reflected in the following amendment to the abstract (l. 41):

Demographic factors which were associated with higher risks of suicidality were female gender, younger age, bad academic results, were an only child, non-participation in school associations, and had an urban household registration.

2. Statistically, I am not quite sure why the authors have chosen to impose a cut-off on the scoring level to determine suicidal ideation or not. I would have thought that it would be more sensible to assess the impact of potential factors on the outcome as a continuous variable. While I do not object to the more simple binary approach used here, it would be good to see a richer approach, treating the SBQ-R as a continuous variable.

Response: We have included Table 2, which treats SBQ-R as a continuous dependent variable in a multiple linear regression model. 

3. Some of the risk factors could also be modelled better. For example, when considering Academic Results: "Academic Results. Those who reported “poor” academic results had the highest level of 231 suicidality mean score at 1.57±0.48, followed by “normal” and “good”, F (2, 11,454) = 62.11,

232 p<0.001. The “good” category scored the lowest among the three categories (p<0.001)." This would seem to suggest that there is a trend effect across the three categories (i.e. Poor > Normal > Good). If that is the case, then please say so, and please consider modelling it.

Response: We have added to the manuscript the observation that there is a trend effect of academic results on suicidal behaviors, as follows (ll. 236-238):

There appears to be a trend effect of the three categories of academic results (i.e. “poor” > “normal” > “good”) on suicidal behaviors.

4. The section on Data Sharing is unclear. The submission states that it is available in the additional material, but that was not made available for review

Response: We have made the data available as a “Supporting Information” file.

5. The Ethics statement in the overview is insufficient; if there was appropriate IRB and individual level consent obtained (as appears to be the case) then this should be stated.

Response: We have added the appropriate IRB approval and the individual level consent statement for this study, as follows (ll. 120-123):

This study received ethics approval from the institutional review board of the Ethics Committee at the School of Public Health, Shandong University (No. 20161103). The participants signed an informed consent form before answering the questionnaire.

Based on this, my recommendation would be to revise and resubmit. I would be happy to review a revised version of the article.

Dr. M. Williams

Imperial College London

---

## [Decision Letter · Decision Letter 1]

6 Jul 2020

PONE-D-20-00103R1

Suicidality among Chinese college students: A cross-sectional study across seven provinces

PLOS ONE

Dear Dr. Siau,

Thank you for submitting your manuscript to PLOS ONE. After careful consideration, we feel that it has merit but does not fully meet PLOS ONE’s publication criteria as it currently stands. Therefore, we invite you to submit a revised version of the manuscript that addresses the points raised during the review process.

We look forward to receiving your revised manuscript.

Kind regards,

Vincenzo De Luca

Academic Editor

PLOS ONE

Reviewers' comments:

Reviewer's Responses to Questions

**Comments to the Author**

1. If the authors have adequately addressed your comments raised in a previous round of review and you feel that this manuscript is now acceptable for publication, you may indicate that here to bypass the “Comments to the Author” section, enter your conflict of interest statement in the “Confidential to Editor” section, and submit your "Accept" recommendation.

Reviewer #1: All comments have been addressed

2. Is the manuscript technically sound, and do the data support the conclusions?

Reviewer #1: Yes

3. Has the statistical analysis been performed appropriately and rigorously? 

Reviewer #1: N/A

4. Have the authors made all data underlying the findings in their manuscript fully available?

Reviewer #1: Yes

5. Is the manuscript presented in an intelligible fashion and written in standard English?

Reviewer #1: Yes

6. Review Comments to the Author

Reviewer #1: Suicidality among Chinese college students: A cross-sectional study across seven provinces – RE-REVIEW

Minor revisions required

The revised version of the paper is well-written and concise. Previous comments were reviewed by the authors and taken account of. It was identified that the response to reviewers were implemented in the latest copy of the study.

Very minor errors were identified and are as stated below.

The following are the line numbers and the recommendations.

1. 79-95 I recommend ‘the collective body of evidence’ to be reformatted. It seems like a run-on sentence that takes a paragraph space on the paper.

2. 119-120 Recommend correcting the wording to ‘approximately a half an hour’.

3. 220 Recommend writing years old (y.o.) before using the abbreviation.

The study as stated previously has been revised and responded to the previous comments left by the reviewers. This version of the study contained small errors that can be corrected as per the discretion of the authors.

After reviewing the paper, I recommend to consider revising and resubmitting.

7. PLOS authors have the option to publish the peer review history of their article (what does this mean?). If published, this will include your full peer review and any attached files.

Reviewer #1: No

---

## [Author Response · Author response to Decision Letter 1]

14 Jul 2020

Dear Editor,

On behalf of my co-authors, I would like to convey our thanks for a further opportunity to revise our manuscript. Please find below our point-to-point response to the reviewer’s comments.

Thank you.

Warm regards

Dr. Siau Ching Sin

Reviewer’s comments:

Minor revisions required

The revised version of the paper is well-written and concise. Previous comments were reviewed by the authors and taken account of. It was identified that the response to reviewers were implemented in the latest copy of the study.

Very minor errors were identified and are as stated below.

The following are the line numbers and the recommendations.

Query #1. 79-95 I recommend ‘the collective body of evidence’ to be reformatted. It seems like a run-on sentence that takes a paragraph space on the paper.

Response: We have revised the paragraph as follows (ll. 79-89):

There is a collective body of evidence on the identification of risk and protective factors for suicide, which includes factors such as life stress and coping style [8], personality factors of impulsivity and aggression [9], and depression [9-12]. In addition, adverse developmental influences or events such as childhood adversity, divorce of parents, loss of a parent, and sexual abuse are also risk factors for increased suicidality [9-12]. Among Chinese college students, factors that have been found to influence suicidality are academic performance, academic stress, occupational future, recent conflicts with classmates, satisfaction with major, and the rupture of romantic relationships [12,13]. Finally, family influences such as parents’ educational level, family income, a history of suicide in the family, and originating from a rural background are also factors that are associated with suicidality among college students [14-19].

Query #2. 119-120 Recommend correcting the wording to ‘approximately a half an hour’.

Response: We have made the correction.

Query #3. 220 Recommend writing years old (y.o.) before using the abbreviation.

Response: We have made the correction.

---

## [Decision Letter · Decision Letter 2]

24 Jul 2020

Suicidality among Chinese college students: A cross-sectional study across seven provinces

PONE-D-20-00103R2

Dear Dr. Siau,

We’re pleased to inform you that your manuscript has been judged scientifically suitable for publication and will be formally accepted for publication once it meets all outstanding technical requirements.

Kind regards,

Vincenzo De Luca

Academic Editor

PLOS ONE

Additional Editor Comments (optional):

Reviewers' comments:

Reviewer's Responses to Questions

**Comments to the Author**

1. If the authors have adequately addressed your comments raised in a previous round of review and you feel that this manuscript is now acceptable for publication, you may indicate that here to bypass the “Comments to the Author” section, enter your conflict of interest statement in the “Confidential to Editor” section, and submit your "Accept" recommendation.

Reviewer #1: All comments have been addressed

2. Is the manuscript technically sound, and do the data support the conclusions?

Reviewer #1: Yes

3. Has the statistical analysis been performed appropriately and rigorously? 

Reviewer #1: I Don't Know

4. Have the authors made all data underlying the findings in their manuscript fully available?

Reviewer #1: Yes

5. Is the manuscript presented in an intelligible fashion and written in standard English?

Reviewer #1: Yes

6. Review Comments to the Author

Reviewer #1: The revised version of the paper is well-written and concise. Previous comments were

reviewed and taken account of.

No further comments.

7. PLOS authors have the option to publish the peer review history of their article (what does this mean?). If published, this will include your full peer review and any attached files.

Reviewer #1: No

---

## [Editor Report · Acceptance letter]

11 Aug 2020

PONE-D-20-00103R2 

Suicidality among Chinese college students: A cross-sectional study across seven provinces 

Dear Dr. Siau:

I'm pleased to inform you that your manuscript has been deemed suitable for publication in PLOS ONE. Congratulations! Your manuscript is now with our production department. 

Kind regards, 

on behalf of

Dr. Vincenzo De Luca 

Academic Editor

PLOS ONE